# Theoretical Insight into the Reaction Mechanism and Kinetics for the Criegee Intermediate of *anti*-PhCHOO with SO_2_

**DOI:** 10.3390/molecules25133041

**Published:** 2020-07-03

**Authors:** Benni Du, Weichao Zhang

**Affiliations:** School of Chemistry and Materials Science, Jiangsu Normal University, Xuzhou, Jiangsu 221116, China; dubn@jsnu.edu.cn

**Keywords:** Criegee intermediate, *anti*-PhCHOO, product distribution, kinetic calculation

## Abstract

In this study, the density functional theory (DFT) and CCSD(T) method have been performed to gain insight into the possible products and detailed reaction mechanism of the Criegee intermediate (CI) of *anti*-PhCHOO with SO_2_ for the first time. The potential energy surfaces (PESs) have been depicted at the UCCSD(T)/6-311++G(d,p)//UB3LYP/6-311++G(d,p) levels of theory with ZPE correction. Two different five-membered ring adducts, viz., *endo* PhCHOOS(O)O (IM1) and *exo* PhCHOOS(O)O (IM2) have been found in the entrance of reaction channels. Both direct and indirect reaction pathways from IM1 and IM2 have been considered for the title reaction. Our calculations show that the formation of PhCHO+SO_3_ (P1) via indirect reaction pathways from IM1 is predominant in all the pathways, and the production of P1 via direct dissociation pathway of IM1 and indirect reaction pathways of IM2 cannot be neglected. Moreover, PhCOOH+SO_2_ (P2) initiated from IM2 is identified as the minor product. According to the kinetic calculation, the total rate constant for the *anti*-PhCHOO+SO_2_ reaction is estimated to be 6.98 × 10^−10^ cm^3^·molecule^−1^·s^−1^ at 298 K.

## 1. Introduction

Criegee intermediate (CI) is an important species formed in the ozonolysis of alkenes, and it is a carbonyl oxide with diradical electronic structure [1]. Since the first direct observation [2] and direct kinetic measurements [3] for the simplest CI, CH_2_OO, recently, much attention has been given to the reactions of CIs in atmospheric chemistry. Due to the significance in ozonolysis mechanism and possessing high reactivity, it has been suggested that CIs can be added as a new type of atmospheric active species in atmospheric oxidation chemistry, besides OH, NO_3_, and O_3_ et al. [4].

When released into the atmosphere, the CIs can react with some atmospheric species, such as H_2_O [3,5,6,7,8,9,10,11,12,13,14], HO_2_ [15,16], NO_2_ [3,10,17,18,19], SO_2_ [3,9,10,11,14,19,20,21,22,23,24,25,26,27,28,29], OH [30] and CH_2_=C(CH_3_)CHO [31] et al., to form corresponding products. The reactions of CIs with SO_2_ in particular play an important role in atmospheric chemistry, due to the formation of SO_3_ and subsequent H_2_SO_4_, which is the main component of aerosols and acid rain [32,33,34,35], and this provides another feasible channel for the formation of H_2_SO_4_ in the atmosphere. 

For the reactions of CIs with SO_2_, many experimental and theoretical studies have been carried out to investigate the kinetics and reaction mechanism for the simplest CH_2_OO [3,5,9,10,11,20,21,22,23,24,25,31], *anti*- or *syn*-CH_3_CHOO and (CH_3_)_2_COO [11,19,21,22,24,26,27,28,30]. Moreover, the reactions of SO_2_ with those CIs produced from czonolysis of limonene, α- pinene, β-pinene [20,29], and styrene [14] etc., have also been studied to investigate the influence of these reactions of CIs+SO_2_ to the formation of sulfuric acid or the secondary aerosols.

In 2017, Díaz-de-Mera et al. [14] studied the ozonolysis of styrene in the presence of SO_2_ at atmospheric pressure and room temperature. The formation of SO_3_ is expected to be major in reaction of Criegee intermediates with SO_2_. They found that lower concentrations of reactants were required in the ozonolysis of styrene with low concentrations of SO_2_ than those required in experiments without SO_2_. Furthermore, under high H_2_O concentrations, the formation of SO_3_ and subsequent H_2_SO_4_ in the smog chamber is inhibited, due to the competitive reactions of CIs with water. However, the rate constants ratio of (2.8 ± 0.7) × 10^−5^ (errors are 2σ ± 20%) for k(H_2_O)/k(SO_2_) illustrates that the reaction of CIs with SO_2_ will be fast under atmospheric conditions.

In the ozonolysis of styrene, both CH_2_OO and C_6_H_5_CHOO (denoted as PhCHOO) will be formed. For the case of CH_2_OO, the reactions with SO_2_ have been studied extensively [3,5,9,10,11,20,21,22,23,24,25,31], both in experiment and theory, while for the reaction of PhCHOO+SO_2_, as far as we know, no corresponding theoretical study has been done. Thus, the main goal of this work is to explore the reaction mechanism and kinetics of *anti*-PhCHOO+SO_2_, give the possible product channels, and compare with the available experimental results. The proposed reaction pathways for the reaction of *anti*-PhCHOO+SO_2_ are presented in Scheme 1.

## 2. Computational Method

The Gaussian 09 package [36] was used to all ab initio calculations. Due to the ability of accurate describing the geometries of transition states and providing some properties that can be comparable in accuracy to higher levels of theory, the hybrid B3LYP [37,38] method has been employed extensively in the CI reaction systems [5,7,13,20,21,24], thus, the geometries of all species involved in the title reaction have been optimized at the B3LYP/6-311++G(d,p) level of theory. Harmonic vibrational frequency calculations were performed at the same level of theory to confirm the character of each stationary point, i.e., all real frequencies for a minimum and only one imaginary frequency for a transition state structure. Moreover, zero-point vibrational energy (ZPE) correction was also obtained from such calculations. Intrinsic reaction coordinate (IRC) [39,40] calculations were also carried out, to verify the connectivity of the transition state structures. To obtain more reliable relative energies, single-point energies of all stationary points at the B3LYP/6-311++G(d,p) geometries were calculated using coupled-cluster theory including single, double, and noniterative triple excitations [CCSD(T)] [41] using the 6-311++G(d,p) basis set. The ZPE corrections with a scale factor of 0.9688 [42] were included in the determination of energy barriers and reaction energies. We noted that the B3LYP/6-311++G(d,p) calculations were unable to locate the reactant complexes RC1, RC2, and transition states TSRC1-1, TSRC2-2 in the entrance channels. However, they could be obtained using the BH&HLYP method [38,43]. Therefore, the stationary points of RC1, RC2, TSRC1-1, and TSRC2-2 were reoptimized at the BH&HLYP/6-311++G(d,p) level. The single-point energy of each optimized geometry at the BH&LYP/6-311++G(d,p) level was recalculated, using the CCSD(T)/6-311++G(d,p) level of theory, with the ZPE correction scaled by 0.9335 [42]. Here, we would like to explain that how the ZPE correction obtained by the DFT method is transformed into the one obtained by the CCSD(T) method. The PESs obtained at 0 K are useful for kinetic analysis, especially for the rate constant calculation at different temperatures. The reliable energies at 0 K of all stationary points can be determined directly by geometry optimization and frequency calculation, using the CCSD(T) method. However, the geometry optimization and frequency calculations at CCSD(T) method are very expensive indeed. In general, the CCSD(T) total energy at 0 K is obtained by adding the DFT-determined ZPE correction to the CCSD(T) single point energy on the basis of DFT geometry optimization. This method is widely used in the PESs construction of reaction systems [44,45]. Moreover, due to the neglection of anharmonicity effects in the theoretical treatment, incomplete incorporation of electron correlation, and the use of finite basis sets, the computed quantum chemical harmonic vibrational frequencies are typically larger than the fundamentals observed experimentally. Therefore, the frequency scale factors are applied to obtain fundamentals and ZPEs [42].

It should be pointed out that, due to the special singlet diradical structure, the closed-shell paradigm is not suitable to CIs. Thus, similar to the reaction of CH_2_OO+SO_2_ [23], the unrestricted broken symmetry approach—mixing the highest occupied molecular orbital (HOMO) and lowest unoccupied molecular orbital (LUMO) option proposed by Noodleman [46]—was used for these species in the geometry optimizations and single-point energies calculations. Moreover, the %TAE[(T)] diagnostic method was used to estimate the magnitude of post-CCSD(T) contributions. It has been shown that %TAE[(T)] diagnostic can provide a useful a priori indicator of the magnitude of the post-CCSD(T) contributions to the total atomization energies (TAEs) [47,48]. According to the %TAE[(T)] values, (%TAE[(T)] = 100 × (TAE[CCSD(T)]−TAE[CCSD])/TAE[CCSD(T)], we can see from Appendix A of Appendix A that all the %TAE[(T)] diagnostic values are small (below 0.02%), which suggests that the post-CCSD(T) contributions should not exceed 0.5 kcal/mol [48], and the application of CCSD(T)/6-311++G(d,p) for this system is suitable.

## 3. Results and Discussion

### 3.1. Reaction Mechanism

In the ozonolysis of styrene, both *syn*- and *anti*- PhCHOO will be formed in the initial steps. The *anti*-PhCHOO species has a smaller steric hindrance than *syn*-PhCHOO in the formation of the corresponding complexes and (or) adducts when drawing near SO_2_, as illustrated in the similar reactions of *anti*-CH_3_CHOO with H_2_O [8] and SO_2_ [26] etc., thus, only the *anti*-PhCHOO conformer is selected as the initial reactant.

The optimized geometries and critical structural parameters for various species, including reactants, reactant complexes (RC), intermediates (IM), transition states (TS), and products associated with the reaction of *anti*-PhCHOO with SO_2_ are shown in Appendix A, Appendix A, and Appendix A of the Appendix A. In Appendix A of the Appendix A, the ZPE, total energies, and relative energies (relative to the isolated species) without and with ZPE corrections of all stationary points calculated at the UCCSD(T)/6-311++G(d,p)//UB3LYP/6-311++G(d,p) levels of theory with ZPE corrections (denoted as UCCSD(T)//UB3LYP) are listed.

The potential energy surfaces (PESs) for the reaction of *anti*-PhCHOO+SO_2_ plotted by the relative energies are shown in Figure 1 and Figure 2, together with the labeling of atoms in the reactant complexes.

Similar to other CIs, the *anti*-PhCHOO produced in the ozonolysis of styrene is highly reactive, and it can be stabilized by collision, followed by the reaction with SO_2_, to form corresponding products.

As shown in Figure 1 and Figure 2, with the approach of SO_2_ from different directions, the reaction can proceed through the formation of two reactant complexes RC1 and RC2 in the entrance channels. Subsequently, with the S atom of SO_2_ bonding to the CI terminal O2 atom and the O3 atom of SO_2_ bonding to the C atom of CI simultaneously, two cyclic adducts *endo* PhCHOOS(O)O (IM1) and *exo* PhCHOOS(O)O (IM2) will be formed via TSRC1-1 and TSRC2-2, respectively. It has been mentioned in the Computational Method section that, at the B3LYP level, no corresponding reactant complexes and transition states have been found in the formation of IM1 and IM2, despite numerous attempts. The structures and energies of RC1, RC2, TSRC1-1, and TSRC2-2 shown in Figure 1 and Figure 2 are obtained at the UBH&HLYP/6-311++G(d,p) and UCCSD(T)/6-311++G(d,p)//UBH&HLYP/6-311++G(d,p), with ZPE corrections (denoted as UCCSD(T)//UBH&HLYP), levels of theory, respectively. The relative energies of RC1 and RC2 are −63.88 and −68.93 kJ/mol, and the corresponding transition states TSRC1-1 and TSRC2-2 lie slightly higher (1.63 kJ/mol for TSRC1-1), or even lower (−0.15 kJ/mol for TSRC2-2) than that of RC1 and RC2, respectively. Obviously, the formation of reactant complexes RC1 and RC2 and the subsequent adducts IM1 and IM2 is (or almost) barrierless. These results can find support from the frontier orbital analysis. As shown in Figure 3, at the UBH&HLYP/6-311++G(d,p) level, the electron density distribution localized on the fragment of –CHOO in the HOMO of *anti*-PhCHOO and the LUMO of SO_2_ is symmetry matching. Furthermore, the energy difference of 0.18539 a.u. between the HOMO of *anti*-PhCHOO and the LUMO of SO_2_ is smaller than 0.34151 a.u. between the HOMO of SO_2_ and the LUMO of *anti*-PhCHOO, thus, the reaction can proceed without barriers via interaction between the HOMO of *anti*-PhCHOO and the LUMO of SO_2_, to form RC1 and RC2.

Comparing the HOMO of RC1 with that of TSRC1-1 shown in Figure 3, we can see that they have similar electronic density distribution and approximate orbital energies, that is to say, from RC1 to TSRC1-1, there is no obvious change of electron density distribution, which results in the formation of IM1 with almost no barrier. The similar instance can also be derived in the formation of IM2, according to the electronic density distribution in the HOMO of RC2 and TSRC2-2.

As depicted in Figure 1 and Figure 2, IM1 and IM2 are a pair of conformational isomers with the S=O double bond pointing to the outside and inside the plane, respectively. IM1 and IM2 lie 146.61 and 143.78 kJ/mol below the initial reactants, and this suggests that they have enough internal energy for the subsequent reactions.

Both isomerization and dissociation pathways of IM1 and IM2 have been considered, and they are discussed as follows.

Firstly, with the simultaneous fracture of O1-O2 and C-O3 bonds initiated from IM1, the products of benzaldehyde (PhCHO)+SO_3_ (P1) will be formed via the transition state TSIM1-P1, by overcoming a barrier of 111.38 kJ/mol.

Secondly, when the H atom shifting from the C1 atom to the adjacent O1 atom, the SO_2_ fragment (O2-S-O4) is also departed from the parent molecule via transition state TSIM1-P2, to produce benzoic acid (PhCOOH). The relative energy of TSIM1-P2 is -14.65 kJ/mol and it is 20.58 kJ/mol higher than that TSIM1-P1. Obviously, the production of P1 is more favorable than that of PhCOOH+SO_2_ (P2), due to its lower energy barrier.

The formation of singlet bisoxy diradical PhC(H)O(O) has also been considered, starting from IM1. However, an attempt to locate the corresponding TS at the B3LYP level was not successful. With the departure of SO_2_ via breaking the O1-O2 and S-O3 bonds, the product of phenyl formate (PhOC(O)H) will be formed via TSIM1-P3 after climbing the barrier height of 125.55 kJ/mol. The IRC analysis of TSIM1-P3 shown in Appendix A verifies that it really connects IM1 and products of PhOC(O)H+SO_2_ (P3). The formation of PhOC(O)H is a new product channel that has not been discussed in other similar CIs+SO_2_ reactions.

In addition to three direct reaction pathways, IM1 can also isomerize to another intermediate IM3 via ring opening transition state TSIM1-3, with a lower barrier of 96.75 kJ/mol. In TSIM1-3, the breaking O1-O2 bond is 1.976 Å. Subsequently, three reaction pathways of IM3 will be open. As shown in Figure 1, the most feasible reaction pathways of IM3 is the formation of P1 via TSIM3-P1. TSIM3-P1 is 54.11 kJ/mol lower in energy than the initial reactants and the barrier height of it is only 32.02 kJ/mol.

As discussed above, the singlet bisoxy diradical PhC(H)O(O) cannot be produced directly from IM1, whereas, via TSIM3-P4, the products of PhC(H)O(O)+SO_2_ (P4) will be formed starting from IM3 by surmounting a barrier of 41.60 kJ/mol. It is worth noting that TSIM3-P4 lies above TSIM3-P1 by about 10 kJ/mol. Moreover, the reaction from IM3 to P4 is endergonic, and the process is reversible, so we can conclude that P4 would never be formed in the reaction.

The third reaction pathway of IM3 is the formation of PhC(O)OSO_2_+H (P5) via C1-H bond rupture transition state TSIM3-P5. The barrier height of TSIM3-P5 is 66.26 kJ/mol, and apparently the formation of P5 is not competitive with that of P1 initiated from IM3, due to it being the highest barrier.

As can be seen from Figure 1, because of the level of the lowest barrier, the formation of P1 through indirect reaction processes, viz. IM1→TSIM1-3→IM3→TS3-P1→P1 is more favorable than that through direct reaction pathway, and is the major product channel, and our calculational result is consistent with the conclusion that the reaction of CI+SO_2_ is surprisingly contributive to the formation of atmospheric H_2_SO_4_ [3]. As for the formation of PhCOOH+SO_2_ (P2), although it is the most favored product channel thermodynamically, the highest barrier height makes this pathway infeasible from IM1.

Similar to IM1, both the direct and indirect reaction pathways of IM2 have been considered, and they are depicted in Figure 2.

The product pathway of IM2 for the formation of P1 has been studied firstly, whereas no right transition state has been located at the B3LYP level. The search for such structure results in the transition state connecting the intermediate formed between *syn*-PhCHOO with SO_2_ and P1.

Starting from IM2, the PhCOOH+SO_2_ (P2) will be formed via TSIM2-P2 with a barrier of 113.32 kJ/mol. As shown in Figure 2, TSIM2-P2 is 15.81 kJ/mol lower in energy than that TSIM1-P2, and this indicates that the formation of P2 via TSIM2-P2 is more feasible. As for the production of P3 from IM2, the transition state TSIM2-P3 involves the highest barrier of 144.32 kJ/mol, thus, the contribution from this product channel can be negligible.

With the fracture of O1-O2 bond, the intermediate IM4 will be formed via TSIM2-4 with a barrier height of 111.38 kJ/mol. IM4 has similar reaction pathways to IM3, viz. the formation of P1, P4, and P5 via TSIM4-P1, TSIM4-P4, and TSIM4-P5, respectively. The relative energies of TSIM4-P1, TSIM4-P4, and TSIM4-P5 are −55.89, −49.29 and −26.46 kJ/mol, respectively. Similar to the reaction from IM3 to P4, the formation of P4 and P5 starting from IM4 can be almost ruled out, since these two pathways are not favorable thermodynamically, and are dynamically compared with the formation of P1 from IM4. As a result, P1 and P2 formed through TSIM4-P1 and TSIM2-P2 are the major and minor products of IM2, respectively.

Comparing Figure 1 with Figure 2, we can see that from both IM1 and IM2, the formation of P1 via indirect reaction pathways is the main product channels. IM1 and IM2 are a pair of isomers with similar structures and closer energies, however, due to their different spatial conformation, the energy of TSIM1-3 is 17.46 kJ/mol lower than that of TSIM2-4, which results in the formation of P1 starting from IM1 to be more feasible, that is to say, IM1 and IM2 have the conformation-dependent reactivity. Moreover, the formation of P2 from IM2 may also have slight contribution to the final product distribution, based on our calculations. In the experiment, besides the dominant SO_3_, no other possible products have been given [14]. Fortunately, our theoretical prediction for the possible product distribution of the title reaction is consistent with the similar reaction of CH_2_OO+SO_2_ [24]. In addition, we can see that during the formation of P1 via indirect reaction pathways of IM1, all the transition states lie below the initial reactants by about 50 kJ/mol, which suggests that the reaction of *anti*-PhCHOO+SO_2_ will be fast. This is attributed to the fact that the effective activation energy Δ*E*_eff_^≠^ (Δ*E*_eff_^≠^ = *E*^TS^ − *E*^Reactants^) [49,50] for the rate-determining step transition state TSIM1-3 is negative. A negative Δ*E*_eff_^≠^ is crucial to the reaction activity and product distribution [51,52]. Nevertheless, this behavior will be testified by further kinetic investigation (see next section).

### 3.2. Kinetic Calculation

The rate constant for the reaction of SO_2_ with the Criegee intermediate can be estimated in terms of the steady state approximation [21] for the IM1 adduct:(1)$$\begin{array}{l} \frac{d\left[\mathrm{IM}1\right]}{dt}=k_{\mathrm{cap}}{[\mathrm{SO}}_{2}][anti-\mathrm{PhCHOO}]-k_{\mathrm{diss},1}\left[\mathrm{IM}1\right]-k_{\mathrm{TSIM}1}{}_{-3}\left[\mathrm{IM}1\right]-k_{\mathrm{TSIM}1-P1}\left[\mathrm{IM}1\right]\\ -k_{\mathrm{TSIM}1-P2}\left[\mathrm{IM}1\right]-k_{\mathrm{TSIM}1-P3}[\mathrm{IM}1]=0 \end{array}$$
(2)$$[\mathrm{IM}1]=\frac{k_{\mathrm{cap}}{[\mathrm{SO}}_{2}][anti-\mathrm{PhCHOO}]}{k_{\mathrm{diss},1}+k_{\mathrm{TSIM}1-3}+k_{\mathrm{TSIM}1-P1}+k_{\mathrm{TSIM}1-P2}+k_{\mathrm{TSIM}1-P3}}$$

Then, the reaction rate starting from IM1 can be represented as:(3)$$\begin{array}{l} r=k_{\mathrm{TSIM}1-3\;}[\mathrm{IM}1]=\frac{k_{\mathrm{TSIM}1-3}\;k_{\mathrm{cap}}{[\mathrm{SO}}_{2}][anti-\mathrm{PhCHOO}]}{k_{\mathrm{diss},1}+k_{\mathrm{TSIM}1-3}+k_{\mathrm{TSIM}1-P1}+k_{\mathrm{TSIM}1-P2}+k_{\mathrm{TSIM}1-P3}}\\ =k_{1}{\;[\mathrm{SO}}_{2}][anti-\mathrm{PhCHOO}] \end{array}$$

Therefore, the rate constant *k*_1_ can be expressed as:(4)$$k_{1}=\frac{k_{\mathrm{TSIM}1-3}\times k_{\mathrm{cap}}}{k_{\mathrm{diss},1}+k_{\mathrm{TSIM}1-3}+k_{\mathrm{TSIM}1-P1}+k_{\mathrm{TSIM}1-P2}+k_{\mathrm{TSIM}1-P3}}$$
where *k*_cap_ represents the capture rate constant between SO_2_ and *anti*-PhCHOO that form the IM1 adduct, *k*_diss,1_ stands for the rate constant for the IM1 adduct dissociates back to the reactants, and *k*_TSIM1-3_, *k*_TSIM1-P1_, *k*_TSIM1-P2_ and *k*_TSIM1-P3_ are the unimolecular rate constants of the dissociation or isomerization channels of the IM1 adduct, and they can be calculated using transition state theory (TST) [53,54,55]:(5)$$k_{uni}=\Gamma \frac{k_{B}T}{h}\frac{Q_{TS}}{Q_{IM1}}exp(-\frac{E_{TS}-E_{IM1}}{RT})$$
where *Q* represents the partition functions of the respective subscripted species, while *E* are the zero-point corrected total energies of the respective subscripted species. *Γ* denotes the Wigner’s tunneling correction [56]. *k_B_*, *h*, *T*, and *R* represent Boltzmann’s constant (1.38 × 10^−23^ J K^−1^), Planck’s constant (6.63 × 10^−34^ J·s), the absolute temperature, and the ideal gas constant (8.314 J mol^−1^ K^−1^), respectively.

As stated above, the formation of IM1 and IM2 is barrierless. The capture rate constant *k*_cap_ can be calculated using the long-range variational TST expression derived by Georgievskii and Klippenstein [57], which can be written as:(6)$$k_{cap}=\frac{C{\left(d_{1}d_{2}\right)}^{2/3}}{\mu^{1/2}T^{1/6}}$$
where *d*_1_ and *d*_2_ are the dipole moments of the SO_2_ and *anti*-PhCHOO, *μ* is the reduced mass of the *anti*-PhCHOO−SO_2_ collision, *T* is the absolute temperature, and *C* is a constant of proportionality whose value is 5.87 [57] for the dipole-dipole interaction between two nonlinear molecules.

The value of *k*_cap_ for the IM1 adduct at 298 K was found to be 6.02 × 10^−10^ cm^3^ molecule^−1^ s^−1^ by using the dipole of *anti*-PhCHOO and SO_2_ (6.5619 D and 2.0269 D at the B3LYP/6-311++G(d,p) level of theory). This value is similar to that obtained in barrierless CH_2_OO-SO_2_ association study [24]. From elementary TST, and using the B3LYP-determined moments of inertia, and the B3LYP-determined vibrational frequencies and the zero-point-corrected CCSD(T) electronic energies, we obtain the following unimolecular rate constants: *k*_diss,1_ = 4.87 × 10^−7^ s^−1^, *k*_TSIM1-3_ = 1.08 × 10^−4^ s^−1^, *k*_TSIM1-P1_ = 4.12×10^−7^ s^−1^, *k*_TSIM1-P2_ = 1.50×10^−10^ s^−1^, *k*_TSIM1-P3_ = 5.25×10^−10^ s^−1^ at 298 K. Thus, the rate constant *k*_1_ is computed to be 5.99 × 10^−10^ cm^3^ molecule^−1^ s^−1^.

By following a similar procedure, the rate constant *k*_2_ for the IM2 adduct can also be obtained as follows:(7)$$k_{2}=\frac{k_{\mathrm{TSIM}2-4}\times k_{\mathrm{cap}}}{k_{\mathrm{diss},2}+k_{\mathrm{TSIM}2-4}+k_{\mathrm{TSIM}2-P2}+k_{\mathrm{TSIM}2-P3}}$$

The unimolecular rate constants beginning with IM2 at 298 K were computed to be *k*_diss,2_ = 1.65 × 10^−6^ s^−1^, *k*_TSIM2-4_ = 3.58 × 10^−7^ s^−1^, *k*_TSIM2-P2_ = 1.73 × 10^−7^ s^−1^, *k*_TSIM2-P3_ =3.29 × 10^−13^ s^−1^, respectively. As a result, the computed value for the rate constant *k*_2_ at 298 K is 9.94 × 10^−11^ cm^3^ molecule^−1^ s^−1^.

Finally, the overall bimolecular rate constant for the *anti*-PhCHOO+SO_2_ reaction was determined to be 6.98 × 10^−10^ cm^3^ molecule^−1^ s^−1^ by adding up the individual bimolecular rate constant *k*_1_ and *k*_2_.

There are no experimental data about the bimolecular rate constant for the *anti*-PhCHOO + SO_2_ reaction. Information on the bimolecular rate constant in the reactions of SO_2_ with Criegee intermediates is available in the case of CH_2_OO and (CH_3_)_2_COO [3,21]. Our calculated value matches within 1 order of magnitude compared with the experimental value of (3.9 ± 0.7) × 10^−11^ cm^3^ molecule^−1^ s^−1^ for the CH_2_OO+SO_2_ reaction recommended by Welz et al. [3]. Moreover, the obtained rate constant presented here agrees well with the computed value of 4 × 10^−10^ cm^3^ molecule^−1^ s^−1^ for both the CH_2_OO+SO_2_ and the (CH_3_)_2_COO+SO_2_ oxidation reactions reported by Kurtén et al. [21].

## 4. Conclusions

Quantum chemical calculations have been performed to characterize the potential energy surface of the *anti*-PhCHOO+SO_2_ reaction at the UCCSD(T)/6-311++G(d,p) level of energy calculations based on UB3LYP/6-311++G(d,p) optimized geometries together with ZPE corrections. Various possible reaction pathways have been probed. The calculated results show that the reaction begins with the formation of two reactant complexes RC1 and RC2 and produces two energy-rich adducts IM1 and IM2 barrierlessly in the process of SO_2_ association with *anti*-PhCHOO from different directions. IM1 and IM2 are a pair of isomers with similar structures and closer energies, however, IM1 shows higher reactivity than that IM2 in the formation of the most favorable product of P1 via indirect reaction pathways, which suggests the conformation-dependent reactivity of *anti*-PhCHOO with SO_2_. P2 is found to be the less competitive product, followed by the almost negligible products of P3, P4 and P5. Based on the quantum chemical calculations and transition state theory, the overall reaction rate constant is predicted to be 6.98 × 10^−10^ cm^3^ molecule^−1^ s^−1^ at 298 K, which is in agreement with the similar CH_2_OO+SO_2_ and (CH_3_)_2_COO+SO_2_ reactions.

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
