# Peer review of "Theoretical Insight into the Reaction Mechanism and Kinetics for the Criegee Intermediate of anti-PhCHOO with SO2"

_molecules, 2020, doi:10.3390/molecules25133041_

Round 1

Reviewer 1 Report

The manuscript brings interesting data on the title topics obtained at a high level of theory. My comments can be summarized as follows:

  1. ZPVE is well defined for stable structures but its physical meaning for transition states is questionable. Therefore I propose to present the energy data both with and without ZPE corrections as well.
  2. The method of Noodleman described in lines 71-72 is known as a ’broken-symmetry’ treatment for DFT methods (with hybrid functionals) in recent literature.
  3. The BHandHLYP method is not mentioned in the ‘Computational Method’ section. If the BHandHLYP data are taken from another literature, its citation is missing.
  4. Equation numbering is missing.
  5. Fig. S2 contains no values in parentheses as indicated in its caption.

Author Response

Comments and Suggestions for Authors

The manuscript brings interesting data on the title topics obtained at a high level of theory. My comments can be summarized as follows:

  1. ZPVE is well defined for stable structures but its physical meaning for transition states is questionable. Therefore I propose to present the energy data both with and without ZPE corrections as well.

Our response: We thank the referee for pointing out this issue. In Table S2 of Supplementary Materials, the energy data both with and without ZPE corrections have been presented. In line 106 of manuscript, the phrase “without and with ZPE corrections” has been added.

  1. The method of Noodleman described in lines 71-72 is known as a ’broken-symmetry’ treatment for DFT methods (with hybrid functionals) in recent literature.

Our response: We thank the referee for pointing out this issue. In line 84 of manuscript, the phrase “the unrestricted broken symmetry approach by mixing the” has been added.

  1. The BHandHLYP method is not mentioned in the ‘Computational Method’ section. If the BHandHLYP data are taken from another literature, its citation is missing.

Our response: We agree with the referee. In lines 76-82 of Computational Method’ section, the BHandHLYP method and corresponding literature are given. Moreover, in lines 126-127, 130-133, the description about BHandHLYP method has been revised.

  1. Equation numbering is missing.

Our response: We thank the referee for pointing out this issue. On pages 8 and 9, the equation numbering has been added.

  1. Fig. S2 contains no values in parentheses as indicated in its caption.

Our response: We thank the referee for pointing out this issue. Due to the geometry parameters of IM1 and IM2 from UBH&HLYP/6-311++G(d,p) level have been given in Fig. S1, so these parameters of IM1 and IM2 are not presented in Fig. S2 and Fig. S3, and the corresponding caption has been deleted.

Reviewer 2 Report

Du and Zhang report a comprehensive DFT/CCSD(T) study about the mechanism of anti-PhCHOO Criegee intermediate reaction with SO2, which is relevant for atmospheric chemistry, especially concerning the formation of SO3 and sulfuric acid as side products. Although the topic is interesting, the main goal of the manuscript may be too specific for a general journal like MOLECULES. Apart from that point, I have minor questions/comments that should be revised before publication elsewhere. Thus, I would recommend publication of the manuscript in a more specialized journal after minor revisions.

Minor corrections reason:

  • The abstract is not clear enough and it is not possible to understand without going deeper into the manuscript. I think that using labels like IM1 or IM2 is not appropriate since the reader should get the essence of the manuscript just by reading the abstract itself. This should be rewritten.
  • The writing is not too clear and should be revised. For example, in page 2, lines 49/50, the authors wrote: “In the ozonolysis of styrene, both CH2OO and C6H5CHOO (denoted as PhCHOO) will be formed in the ozonolysis of styrene
  • A Figure/scheme in the introduction, including the reaction studied and its conditions, would help to follow and understand the paper.
  • According to Fig1 and Fig 2, the authors proposed that P1 and P4 can be formed as major and minor products, respectively. However, according the energies reports, P4 would never be formed in the reaction, since the reaction from IM4 to P4 is endergonic, and the process is reversible, and so it would end up always in P1. The authors should comment on that.
  • In page 6, the authors said that the reaction should be fast because all the TSs lie below the initial reactant. However, the reaction rate does not depend on that fact, but in the highest difference between the most stable intermediate and the highest transition state of the corresponding reaction channel. This should be commented.
  • Add xyz to the supporting information.

Author Response

Comments and Suggestions for Authors

Du and Zhang report a comprehensive DFT/CCSD(T) study about the mechanism of anti-PhCHOO Criegee intermediate reaction with SO2, which is relevant for atmospheric chemistry, especially concerning the formation of SO3 and sulfuric acid as side products. Although the topic is interesting, the main goal of the manuscript may be too specific for a general journal like MOLECULES. Apart from that point, I have minor questions/comments that should be revised before publication elsewhere. Thus, I would recommend publication of the manuscript in a more specialized journal after minor revisions.

Minor corrections reason:

  • The abstract is not clear enough and it is not possible to understand without going deeper into the manuscript. I think that using labels like IM1 or IM2 is not appropriate since the reader should get the essence of the manuscript just by reading the abstract itself. This should be rewritten.

Our response: We thank the referee for pointing out this issue. In line 15, IM1 and IM2 are redefined as endo PhCHOOS(O)O (IM1) and exo PhCHOOS(O)O (IM2). The abstract has been rewritten.

  • The writing is not too clear and should be revised. For example, in page 2, lines 49/50, the authors wrote: “In the ozonolysis of styrene,both CH2OO and C6H5CHOO (denoted as PhCHOO) will be formed in the ozonolysis of styrene

Our response: We thank the referee for pointing out this issue. In lines 53 and 54, the sentence of “In the ozonolysis of styrene, both CH2OO and C6H5CHOO (denoted as PhCHOO) will be formed in the ozonolysis of styrene” has been revised as “In the ozonolysis of styrene, both CH2OO and C6H5CHOO (denoted as PhCHOO) will be formed”.

On page 3, line 103, the phrase of “reaction complexes” has been revised as “reactant complexes”.

On page 7, lines 221-222, the phrase of “P2 may also have some contribution” has been revised as “the formation of P2 from IM2 may also have slight contribution”.

On page 9, line 299, the phrase of “various species in involved” has been revised as “various species involved”

On page 7, line 208, the phrase of “the formations of P1, P4…” has been revised as “the formation of P1, P4…”

  • A Figure/scheme in the introduction, including the reaction studied and its conditions, would help to follow and understand the paper.

Our response: We thank the referee for pointing out this issue. On page 2, the Scheme “proposed mechanisms for the anti-PhCHOO+SO2 reactions” has been added.

  • According to Fig1 and Fig 2, the authors proposed that P1 and P4 can be formed as major and minor products, respectively. However, according the energies reports, P4 would never be formed in the reaction, since the reaction from IM4 to P4 is endergonic, and the process is reversible, and so it would end up always in P1. The authors should comment on that.

Our response: We agree with the referee. The formation of P4 starting from IM3 or IM4 can be almost ruled out since these two pathways are not favorable thermodynamically and dynamically. On page 7, lines 182-184, line 187, lines 193-194, lines 210-215, lines 217 and 220, the corresponding description about P4 has been rewritten. Moreover, in Abstract and Conclusion, the description about P4 also has been revised.

  • In page 6, the authors said that the reaction should be fast because all the TSs lie below the initial reactant. However, the reaction rate does not depend on that fact, but in the highest difference between the most stable intermediate and the highest transition state of the corresponding reaction channel. This should be commented.

Our response: We thank the referee for pointing out this issue. On pages 7 and 8, lines 227-231, the corresponding explanation is given as follows: This attributes to the fact that the effective activation energy ΔEeffEeff=ETS-EReactants) [47,48] for the rate-determining step transition state TSIM1-3 is negative. A negative ΔEeff is crucial to the reaction activity and product distribution [49,50]. Nevertheless, this behavior will be testified by further kinetic investigation (see next section).

  • Add xyz to the supporting information.

Our response: We thank the referee for pointing out this issue. The Cartesian coordinates have been added in the Supplementary Materials.

Round 2

Reviewer 1 Report

The manuscript brings interesting data on the title topics obtained at a high level of theory. My comments can be summarized as follows:

  1. The ZPE correction evaluation is problematic for the calculations of the CCSD(T)//B3LYP and CCSD(T)//BHandHLYP types. In general, vibrational analysis is of physical meaning for the same quantum-chemical method only which has been used for geometry optimization as well. Authors must explain how they transformed the ZPE correction obtained by the DFT method into the one obtained by the CCSD(T) method (scaling factors?). If the supposition of the same optimal structures obtained by both methods has been used – this must be proven (vanishing gradients ?).
  2. Ad Figs. 1 and 2 captions – it must be pecified that the potential energy profiles are with ZPE corrections (as mentioned in the main text).

Author Response

Comments and Suggestions for Authors

The manuscript brings interesting data on the title topics obtained at a high level of theory. My comments can be summarized as follows:

  1. The ZPE correction evaluation is problematic for the calculations of the CCSD(T)//B3LYP and CCSD(T)//BHandHLYP types. In general, vibrational analysis is of physical meaning for the same quantum-chemical method only which has been used for geometry optimization as well. Authors must explain how they transformed the ZPE correction obtained by the DFT method into the one obtained by the CCSD(T) method (scaling factors?). If the supposition of the same optimal structures obtained by both methods has been used – this must be proven (vanishing gradients ?).

Our response: We thank the referee for pointing out this issue. On page 3, lines 82-90, we explained that how the ZPE correction obtained by the DFT method was transformed into the one obtained by the CCSD(T) method. Moreover, the reason for the ZPE scale factors application has also been given on page 3, lines 91-94.

  1. Ad Figs. 1 and 2 captions – it must be pecified that the potential energy profiles are with ZPE corrections (as mentioned in the main text).

Our response: We thank the referee for pointing out this issue. The phrase of “with ZPE corrections” has been added to the Figs. 1 and 2 captions.

We are greatly thankful to the reviewer for his/her valuable comments.

Reviewer 2 Report

The authors have addressed my comments, so I would recommend publication without further revisions.

Author Response

Comments and Suggestions for Authors

The authors have addressed my comments, so I would recommend publication without further revisions.

Our response: We thank Reviewer #2 for his/her positive comments on our manuscript.